# Tumor-Associated Macrophages Affect the Tumor Microenvironment and Radioresistance via the Upregulation of CXCL6/CXCR2 in Hepatocellular Carcinoma

**DOI:** 10.3390/biomedicines11072081

**Published:** 2023-07-24

**Authors:** Hsin-Lun Lee, Yi-Chieh Tsai, Narpati Wesa Pikatan, Chi-Tai Yeh, Vijesh Kumar Yadav, Ming-Yao Chen, Jo-Ting Tsai

**Affiliations:** 1Department of Radiology, School of Medicine, College of Medicine, Taipei Medical University, Taipei 11031, Taiwan; 132018@h.tmu.edu.tw; 2Department of Radiation Oncology, Taipei Medical University Hospital, Taipei 11031, Taiwan; 3The PhD Program for Translational Medicine, College of Medical Science and Technology, Taipei Medical University and Academia Sinica, Taipei 11031, Taiwan; 4Department of Radiation Oncology, Cancer Center, Taipei Medical University—Shuang Ho Hospital, New Taipei City 23561, Taiwan; 11445@s.tmu.edu.tw; 5Division of Urology, Department of Surgery, Faculty of Medicine, Universitas Gadjah Mada, Yogyakarta 55281, Indonesia; narpatiwp@gmail.com; 6Department of Medical Research and Education, Taipei Medical University—Shuang Ho Hospital, New Taipei City 23561, Taiwan; ctyeh@s.tmu.edu.tw; 7Continuing Education Program of Food Biotechnology Applications, College of Science and Engineering, National Taitung University, Taitung 95092, Taiwan; 8Division of Gastroenterology and Hepatology, Department of Internal Medicine, School of Medicine, College of Medicine, Taipei Medical University, Taipei 11031, Taiwan; vijeshp2@gmail.com; 9Division of Gastroenterology and Hepatology, Department of Internal Medicine, Taipei Medical University—Shuang Ho Hospital, New Taipei City 23561, Taiwan

**Keywords:** hepatocellular carcinoma, tumor microenvironment, tumor-associated macrophages

## Abstract

Background: Hepatocellular carcinoma is the sixth most diagnosed malignancy and the fourth most common cause of cancer-related mortality globally. Despite progress in the treatment of liver cancer, nonsurgical treatments remain unsatisfactory, and only 15% of early-stage cases are surgically operable. Radiotherapy (RT) is a non-surgical treatment option for liver cancer when other traditional treatment methods are ineffective. However, RT has certain limitations, including eliciting poor therapeutic effects in patients with advanced and recurrent tumors. Tumor-associated macrophages (TAMs) are major inflammatory cells in the tumor microenvironment that are key to tumor development, angiogenesis, invasion, and metastasis, and they play an essential role in RT responses. Methods: We used big data analysis to determine the potential of targeting CXCL6/CXCR2. We enrolled 50 patients with liver cancer who received RT at our hospital. Tumor tissue samples were examined for any relationship between CXCL6/CXCR2 activity and patient prognosis. Using a cell coculture system (Transwell), we cocultured Huh7 liver cancer cells and THP-1 monocytes with and without CXCL6/CXCR2 small interfering RNA for 72 h. Results: The overexpression of CXCL6/CXCR2 was highly correlated with mortality. Our tissue study indicated a positive correlation between CXCL6/CXCR2 and M2-TAMs subsets. The coculture study demonstrated that THP-1 monocytes can secrete CXCL6, which acts on the CXCR2 receptor on the surface of Huh7 cells and activates IFN-g/p38 MAPK/NF-κB signals to promote the epithelial–mesenchymal transition and radio-resistance. Conclusions: Modulating the TAM/CXCL6/CXCR2 tumor immune signaling axis may be a new treatment strategy for the effective eradication of radiotherapy-resistant hepatocellular carcinoma cells.

## 1. Introduction

Hepatocellular carcinoma (HCC) is a highly aggressive cancer often resulting in metastasis and disease recurrence post-treatment. This trait is driven by multiple factors, such as cellular and non-cellular elements of the tumor microenvironment (TME). Among these, chemokines, key macromolecular regulators of cell migration and immune response, significantly influence cellular behavior relative to external stimuli [1].

Studies are increasingly highlighting the potential role of chemokines in fostering tumor growth, angiogenesis, and metastasis. They achieve this by cultivating a tumor microenvironment that is abundant in oxygen and nutrients, both of which are elements that are essential for tumor metastasis [1]. Chemokines’ varied functions indicate that they serve as essential links between cancer cells and the tumor microenvironment (TME), playing a significant role in disease progression and metastasis. Notably, metastasis is the leading cause of cancer-related mortality in hepatocellular carcinoma (HCC) patients [2]. A wealth of evidence underscores a strong correlation between chemokine and chemokine receptor signaling, their expression in both primary and metastatic cancers, and patient prognosis [3].

Chemokines facilitate metastasis by directly supporting cell survival, angiogenesis, and invasion and by modulating the reorganization of pre-metastasis TME and the anticancer immune response [3]. The TME is a niche for interactions between cancer cells and the immune system of the host, especially because the TME is infiltrated by various subsets of immune cells in response to interactions between chemokines and their receptors [3,4]. These different immune cell subsets exert different effects on disease progression, treatment response, and clinical outcomes [4]; thus, because the intratumoral infiltration of these immune cells is regulated by chemokines, chemokines may affect anticancer immunity and the treatment response in patients with HCC. Thus, by targeting the chemokine–chemokine receptor cascade, altering cancer biology and the associated immunological phenotype is possible, resulting in enhanced anticancer immune responses [1,3,4].

Among the leukocytes, macrophages are the most abundant and influential in tumor development [5]. Typically referred to as tumor-associated macrophages (TAMs), these macrophages may either infiltrate or reside within the TME [6]. The presence of TAMs in tumor tissue is often correlated with a worse prognosis in cancer patients across various types, implying that these macrophages may contribute significantly to disease progression [7].

The CXCR2 receptor and its CXCL6 ligand are gaining significant attention within the scientific community due to their critical roles in immune regulation and inflammatory responses [8]. They exert a considerable influence on the pathogenesis of various diseases, including cancer, inflammatory disorders, and cardiovascular conditions [9]. Notably, as part of their multiple functions, these chemotactic entities induce the mobilization of professional phagocytes such as neutrophils, dendritic cells, and macrophages to inflammation sites; this facilitates efficient engulfment and the neutralization of foreign pathogens [3,4,7].

It has been reported that CXCL6, CXCR1, and CXCR2 are aberrantly expressed both at the transcript and protein levels in several osteosarcoma cell lines and that the ectopic expression of CXCL6 enhances tumor growth, metastasis to the lungs, and the activation of Wnt/β-catenin and PI3K/AKT pathways in immunocompromised mice in vivo, suggesting the crucial role of the CXCL6/CXCR1 axis in facilitating the proliferation and metastasis of cancer cells [10].

CXCL6, a member of the elastin-like recombinant (ELR)+CXCL family, regulates its downstream targets or associated pathways by coupling with CXCR1 and CXCR2 [10]. Moreover, CXCL6 could act as a prognostic indicator for the progression of non-alcoholic steatohepatitis (NASH) during hepatic inflammation in non-alcoholic fatty liver disease (NAFLD) patients [11]. The dysregulated expression of CXCL6, noted in cases of HCC, emphasizes its relevance in HCC development. Prior research has shown that the up-regulated expression of CXCR1 or CXCR2 ligands, specifically CXCL1, CXCL2, and CXCL8, is linked to chemoresistance and suspected to hamper the anti-cancer immune response [12].

Hence, a detailed exploration of the interaction between CXCR2 and CXCL6 could stimulate progress in the development of treatment approaches, improving patient prognosis in various diseases. This prompted us to examine the effects of the CXCL6/CXCR2 interaction on the TME and immune responses in HCC chemoresistance and progression.

## 2. Materials and Methods

### 2.1. HCC Cell Culture

This study used a human HCC Huh7 cell line (cat # ABC-TC0437) obtained from AcceGen Biotech (Santa Monica, CA, USA), and the cells were cultured in RPMI 1640 (Thermo Fisher Scientific, Waltham, MA, USA) supplemented with 10% fetal bovine serum (Thermo Fisher Scientific) and 100 U/mL of penicillin–streptomycin (CellGro, Lincoln, NE, USA). The Huh7 cell line was established from a patient with well-differentiated HCC and exhibited constitutive p53 mutation. This cell line was already available in our laboratory.

### 2.2. The Cancer Genome Atlas and Gene Expression Omnibus Database Extraction

We used a dataset of patients with HCC from the Cancer Genome Atlas (TCGA) and Gene Expression Omnibus (GEO) portals. Formatted data containing the necessary clinical information were extracted and integrated with corresponding microarray data, including data regarding HCC tumors and normal liver tissues. Extracted clinical profiles included patient age, sex, ethnicity, follow-up duration (days), endpoint/event, method of an initial confirmed diagnosis, histological type, pathological stage, and grade.

### 2.3. Cell Viability Assay

The evaluation of cell viability was carried out on both adherent and spheroid HCC cells following a 30 min irradiation at a predetermined dosage. Additionally, calculations were carried out for combination indices to ascertain the nature of drug interactions—whether they are synergistic, additive, or antagonistic. To measure cell viability, we employed a Sulforhodamine B (SRB) assay. Cells were seeded at a density of 3000 cells/well in 96-well plates. After 24 h, the adherent cells were randomly assigned to either control or treatment groups, and 100 µL of 10% trichloroacetic acid (TCA) was applied to each well. Following an hour-long incubation at 4 °C, the TCA was removed, and 100 µL of SRB reagent was introduced to each well, succeeded by another hour of incubation at room temperature. The cells were then washed with 1% acetic acid. After drying the plates for 20 min at 60 °C, each well received 200 µL of 20 nM Tris buffer. Absorbance readings were carried out at 565 nm using a spectrophotometer. Cell viability, expressed as a percentage, was then computed using absorbance measurements, contrasting the treatment and control groups.

### 2.4. Western Blotting

Western blotting was performed as per the standard protocol using equal amounts of proteins. The antibodies used were those against CXCL6 (#ab198505), CXCR2 (#ab89254), CD163 (#ab182422), vimentin (#ab8979), IFN-g (#ab218426), NF-κB p65 (#ab86299), p-NF-κB p65 (#ab76302), p38-MAPK (#ab32142), p-p38-MAPK (#ab38238), and p-Chk2 (#ab32148) from Abcam and Rad50 (#sc-74460), Chk2 (#sc-5278), MLH3 (#sc-25313), MSH3 (#sc-271080), g-H2AX (#sc-517336), ERCC2 (#sc-271206), and GAPDH (sc-32233) from Santa Cruz Biotechnology (Santa Cruz, CA, USA). The list of used antibodies and dilution is shown in Appendix A.

### 2.5. Immunohistochemistry Staining

Immunohistochemistry (IHC) staining analyses were performed on formalin-fixed paraffin-embedded sections by using antibodies against CXCL6 (#ab198505), CXCR2 (#ab89254), and CD163 (#ab182422). Staining positivity was measured as the percentage of positive cells and the intensity of staining. The positivity cutoff was the reactivity of HCC cells with antibodies of ≥10%. All staining results were reviewed and quantified independently by two pathologists blinded to the clinical outcome. The study was approved by the Joint Institutional Review Board (JIRB) of the Taipei Medical University—Shuang Ho Hospital (Approval no.: JIRB N202203097).

### 2.6. Immunofluorescence (IFC) Staining

Human HCC tissue sections were baked at 62 °C for 1 h and treated with alcohol serially for rehydration; antigen retrieval was then performed by microwaving the tissue in 0.01 mol/L sodium citrate for 20 min. The sections were blocked with serum for 30 min and incubated with primary antibodies in phosphate-buffered saline at 4 °C overnight and then incubated with Cy5-labeled secondary antibodies for 1 h. The nuclei were stained with DAPI. The stained sections were mounted with an antifade solution and then imaged using microscopy.

### 2.7. RNA Extraction and Quantitative Real-Time Reverse Transcription Polymerase Chain Reaction (RT-PCR)

Total RNA was extracted using the TRIzol reagent (Thermo Fisher Scientific) as per the manufacturer’s instructions. A spectrophotometer (NanoDrop 2000, Thermo Fisher Scientific) was used for the quantification of the RNA concentration. We performed the reverse transcription reaction using 2 μg of total RNA with a cDNA first-strand synthesis kit (Quantscript RT Kit, Tiangen Biotechnology, Beijing, China). The mRNA expression level was determined using an SYBR Green qPCR kit (QuantiTect, Qiagen, Hilden, Germany) and a real-time PCR System (LightCycler 480, Roche, Basel, Switzerland). The inhibition of CXCL6 or CXCR2 expression achieved using si-RNA (Sigma, St. Louis, MO, USA) was transfected into the cells using lipofectamine 2000 reagents (Invitrogen, Waltham, MA, USA) according to the vendor’s instructions.

### 2.8. Statistical Analysis

All statistical analyses were performed using GraphPad Prism 6.0 (San Diego, CA, USA) and SPSS Statistics version 25.0 (IBM, Armonk, NY, USA). Pearson’s *X*^2^ test was used to assess the association between pairs of categorical variables. Within-group differences in continuous variables were compared using paired Student’s *t*-tests, one-way analysis of variance, and multivariate analysis. All experiments were performed in triplicates. Survival rates between the groups were compared using Kaplan-Meier curves. A *p* value of <0.05 indicated significance.

## 3. Results

### 3.1. High Expression of CXCL6 and CXCR2 Was Associated with Hepatogenesis and Reduced the Overall Survival of Patients with HCC

The results of a preliminary probe in an HCC cohort (*n* = 57) were obtained by Mas et al. [13]. using the Oncomine platform, and the results demonstrated that patients with HCC exhibited a higher expression of CXCL1 (1.35-fold, *p* = 0.003) and CXCL6 (3.75-fold, *p* = 3.15 × 10^−7^) than their normal counterparts (Figure 1A). A similar pattern was observed; the expression of CXCR1 was suppressed in HCC samples compared to their normal part (−1.33-fold, *p* = 1.00), and the expression of CXCR2 was induced in the HCC samples compared to their normal tissue (1.137-fold, *p* = 0.004) (Figure 1B). These results suggest the roles of CXCL1, CXCL6, and CXCR2 in the development of HCC. To determine the functional relevance of this finding, we reanalyzed Hoshida Golub Liver GSE10143 (*n* = 162) data and demonstrated that the high expression of CXCR2 and CXCL6 was associated with worse overall survival in patients with liver cancer (Figure 1C).

### 3.2. Strong Link between CXCL6 and CXCR2 Expression, M2 Macrophage Infiltration, and Unfavorable Prognosis in Hepatocellular Carcinoma Progression

Considering the potential significance of CXCL6 overexpression and its CXCR2 receptor in the tumor microenvironment (TME), with implications for HCC development and progression, we conducted staining procedures on clinical samples derived from our hospital’s hepatocellular carcinoma (HCC) cohort (*n* = 50). In line with our previous findings, we noted that patients with HCC demonstrated a higher expression of CXCL6 and CXCR2 compared to individuals without the disease in the control group (Figure 2A). Our observations revealed a robust positive correlation between the expression of CXCL6, CXCR2, and the CD163 macrophage-associated antigen, as evidenced in Figure 2A,B. These findings underscore the significance of dysregulated CXCL6/CXCR2 signaling in the functionality of tumor-associated macrophages (TAMs). The bar plot, which quantifies the percentage of CXCL6/CXCR2 expression in M2 macrophages with either high or low CD163 expression, illustrates this relationship further. Specifically, a high CD163 expression is strongly associated with elevated expression levels of both CXCL6 (82.86%) and CXCR2 (74.29%). Conversely, a lower expression of CD163 corresponds to the decreased expression levels of CXCL6 (26.67%) and CXCR2 (40.4%). Furthermore, survival analysis demonstrated that similarly to the expression of CXCL6 and CXCR2, the high expression of CD163 negatively affected the overall survival of patients with HCC (Figure 2C). A similar survival profile was found in individuals with a high expression of CD206, a marker of the M2 phenotype (Figure 2C). Consistent with this finding, our univariate and multivariate analyses demonstrated that CXCL6 and CXCR2 expression were strong factors influencing overall survival in our cohort of patients with HCC.

### 3.3. TAMs Promoted HCC Cell Proliferation and Migration by Activating CXCR2/IFN-g/p38 MAPK/NF-κB Signaling

To better understand the association between CXCL6/CXCR2 signaling and the tumor-promoting anti-inflammatory M2 phenotype of TAMs, we used an in vitro coculture system consisting of Huh7 cells and macrophages. Using a double-layered culture method, we first incubated both types of cells separately. After the cells reached confluence, Huh7 cells were cultured in the upper layer, and macrophage cells were cultured in the lower layer. The cells migrated and were then quantified (Figure 3A). Our preliminary result demonstrated that compared with the singly cultured Huh7 cells, Huh7 cells cocultured with macrophages exhibited enhanced proliferation, which was reversed by the transfection of macrophages with si-CXCR2 or si-CXCL6 (Figure 3B). The effectiveness of si-RNA transfection on the cells has been depicted in Appendix A. Similarly, migration was also enhanced in CXCL6-cocultured cells but was suppressed in the macrophages transfected with si-CXCR2 or si-CXCL6 (Figure 3C). Western blot data demonstrated that transfection with si-CXCL6 or si-CXCR2 caused the modulation of CXCR2 and CD163 expression and concurrently downregulated IFG-γ, p-NF-κB, and p-p38 MAPK expression, whereas the reverse was observed with the introduction of the CXCL6 agnostic into the system (Figure 3D). A similar expression profile was obtained when cells were treated with a CXCL6 agonist (Figure 3E).

### 3.4. CXCL6 Conferred Radioresistance via the Enhancement of DNA Damage Repair

We demonstrated that coculturing Huh7 cells with macrophages enhanced the viability of Huh7 cells exposed to radiotherapy compared with the control cells or those pretreated with a CXCL6 or CXCR2 antagonist (Figure 4A). We used AZD5069, an innovative, powerful, selective, and practically applicable small-molecule inhibitor of CXCR2, as a clinically focused strategy to enhance the efficacy of radiotherapy and immunotherapy for HCC. The enhanced survival of irradiated Huh7 cells was associated with the upregulated expression of DNA repair markers γ-H2AXa, Rad50, p-Chk2, MLH3, MSH3, and ERCC2 in the macrophage cocultured cells compared with the control cells (Figure 4B). Figure 4C represents the schematic description of the aforementioned process. Furthermore, the Kaplan–Meier (KM) plot (Figure 4D) illustrated the correlation between the expression of the H2AFX gene with patient survival rates and the induced expression of H2AFX associated with poor liver cancer patient survival.

### 3.5. Enhanced CXCL6/CXCR2 Signaling Was Associated with Low-Dose (0.5-Gy) Radiation-Induced Epithelial–Mesenchymal Transition (EMT) of HCC Cells

In addition to the preliminary findings, we observed that low-dose radiation significantly upregulated the expression of the CXCL6 protein and mRNA (Figure 5A,B) and CXCR2 mRNA (Figure 5C). Furthermore, we observed that the CXCR2 inhibitor, AZD5069, significantly diminished the expression of vimentin after the application of low-dose radiotherapy (Figure 5D).

## 4. Discussion

HCC is the sixth most diagnosed malignancy and the fourth most common cause of cancer-related death worldwide. Despite advances in liver cancer treatment, nonsurgical treatment remains inadequate, and only 15% of early-stage cases are surgically operable. Radiotherapy (RT) is a non-surgical treatment option for liver cancer when other traditional treatment methods are ineffective. However, in the management of HCC, radioresistance remains a problem that requires addressing. To overcome this difficulty, this study evaluated the role of TAMs in influencing the TME and causing radioresistance in HCC via the regulation of chemokine expression.

Chemokines are a class of secreted proteins that participate in various physiological and pathological processes, such as oncogenesis, immunity, and homeostatic control [1]. CXCL1+, CXCL2, CXCL3, CXCL5, CXCL6, CXCL7, and CXCL8 families have been demonstrated to be proangiogenic. In addition, ELR+CXCLs aid in the development of cancer [14]. In a previous study, CXCR2 was demonstrated to promote HCC immune evasion by regulating PD-L1 [15]. The same study also demonstrated that CXCR2 knock-out cells activated macrophages into the M1 phenotype, thus increasing the M1/M2 ratio [15]. M2-like TAMs are critical in the progression and therapy of cancers. Accordingly, Granulocyte chemotactic protein 2 (GCP-2)/CXCL6 is a CXC chemokine that macro-phages, along with epithelial and mesenchymal cells, express during inflammation [16]. This chemokine, a small, inducible molecule with chemotactic properties, is pervasive and plays a notable role in both acute and chronic inflammation [17]. Among such chemokines, soluble CXCL6 emerges as a vital inflammatory cytokine. It facilitates the assembly of inflammatory cells at inflammation sites by engaging with CXCR1 and CXCR2 receptors [18]. Dysregulated CXCL6 expression has been detected in cases of inflammatory bowel disease and periodontitis [19]. Additionally, it’s found that CXCL6 can hasten the course of lung fibrosis [20], and its concentration markedly increases in patients diagnosed with idiopathic pulmonary fibrosis [20]. Notably, CXCL6 plays a pivotal role in liver fibrosis evolution by provoking the release of TGF-β from Kupffer cells (KCs), leading to the activation of the hepatic stellate cell (HSC) line [21]. CXCL6’s association with the severity of hepatic inflammation in NAFLD patients suggests it could serve as a predictive marker for NASH progression [11].

Accordingly, CXCL6, which can be released by various tumor cells, including HCC cells, is essential for the progression of malignancies. Farha et al. reported that an aggressive HCC phenotype is characterized by discrete immune cell clusters with a macrophage preponderance, which are clinically associated with a poor prognosis and sorafenib response and are molecularly defined using angiogenic gene enrichment [22].

An initial study into the HCC cohort by Mas et al., consisting of 57 participants and utilizing the Oncomine platform, found increased levels of CXCL1 expression in HCC patients. In these patients, high expression of CXCL6 and CXCR2 closely correlated with tumor infiltration by M2 macrophages, disease progression, and poor prognosis. Our study, using clinical samples from our hospital’s HCC cohort, confirmed these findings: HCC patients displayed elevated CXCL6 and CXCR2 expression com-pared to their healthy counterparts. These results lend credence to the hypothesis that overexpression of chemokine CXCL6 and its receptor CXCR2 in the tumor microenvironment is crucial for HCC development and progression, suggesting that dysregulated CXCL6/CXCR2 signaling facilitates tumour-associated macrophage (TAM) activation. Moreover, our survival study showed that, similarly to CXCL6 and CXCR2, heightened CD163 expression had a negative impact on the overall survival of HCC patients.

Depending on whether the amino acid triplet ELR is present in the protein sequence of a cytokine of the C-X-C motif chemokine ligand (CXCL) family, these cytokines exert various biological effects. ELR+CXCL exhibits proangiogenic properties, whereas ELR-CXCL exhibits antiangiogenic properties [14]. Granulocyte Chemotactic Protein 2 (GCP2), or CXCL6, was first discovered in MG-63, an osteosarcoma cell line, by Proost et al. [23]. CXCL6 is an ELR+CXCL. CXCL6 modulates its downstream pathways by interacting with CXCR1 and CXCR2, which are both ELR+CXCLs, similar to CXCL8 [24].

According to our early findings, Huh7 cells cocultured with macrophages demonstrated greater proliferation than those grown alone, which was inhibited by the transfection of macrophages with si-CXCR2 or si-CXCL6. By triggering CXCR2/IFN-g/p38 MAPK/NF-κB signaling, TAMs lead to increased HCC cell proliferation and migration. According to Western blotting results, transfection with si-CXCL6 or si-CXCR2 caused the downregulation of CXCL6, CD163, IFN-g, p-NF-B, and p-p38 MAPK expression. CXCL6 increased DNA damage repair, which enhanced radioresistance. Furthermore, we demonstrated that compared with the control or cells pretreated with CXCL6 or CXCR2 antibodies, Huh7 cells cocultured with macrophages had superior viability when subjected to radiation. Consistently with the preliminary results, we identified that low-dose radiation markedly increased the expression levels of CXCL6 protein and mRNA as well as CXCR2 mRNA (Figure 5A,B). However, we combined low-dose irradiation and AZD5069, which is the novel inhibitor of CXCR2. AZD5069 was designed to be selective for CXCR2. This medication was approximately 100 times more effective against CXCR2 than against CXCR1 when evaluated in a comparable setup with recombinant CXCR1 expression. This medication has exhibited safety and acceptability in individuals with advanced cancers and chronic obstructive pulmonary disease (COPD). AZD5069 has also exhibited rapid absorption in healthy individuals [25,26,27]. This particular experiment demonstrated that the increased expression of CXCL6 did not influence vimentin expression in the absence of CXCR2, emphasizing the role of the CXCL6/CXCR2 axis in HCC EMT.

Adaptive responses, cell proliferation, angiogenesis, and extracellular matrix re-modelling—changes that all contribute to the development of a premetastatic niche—are critical events required for cancer progression and are orchestrated by TAMs [28]. TAMs must be further subclassified because their polarization affects their behavior. Fundamentally, macrophages are divided into the proinflammatory, antifibrotic M1 subtype (which is activated by LPS, TNF, and IFN-g) and the proinflammatory, profibrotic M2 subtype (which is triggered by interleukin (IL)4 and IL13) [29]. Emerging classification paradigms depict that TAMs exist as a continuum of multiple subtypes or even as a mixed phenotype that is neither M1 nor M2, because of the dynamic nature of the TME and the various stimuli within it. TAMs generate cathepsins, matrix metalloproteinases, and angiogenic growth factors, all of which support the angiogenesis of cancer. Moreover, TAMs promote EMT, which aids in tumor spreading [28,30].

We hypothesized that similarly to CXCL2 or CXCL8, CXCL6 might control the EMT process in HCC cells through the CXCR2 axis. According to Cheng et al., CXCL8 accelerated colorectal cancer development by triggering EMT. Additionally, they demonstrated that the administration of recombinant human CXCL6 produced the opposite effect compared to that exerted by treatment with the anti-CXCL6 antibody, which decreased N-cadherin and snail levels and substantially enhanced E-cadherin levels. Another study investigating the role of CXCL2/CXCR2 identified that short hairpin RNA inhibition of CXCL2 reduced the expression of EMT markers in colon cancer cells [22,31]. The findings corroborate our proposed theory that the induction of the epithelial–mesenchymal transition (EMT) by CXCL6 renders hepatocellular carcinoma (HCC) cells more resistant to radiation.

Our schematic abstract of Figure 6 shown below demonstrated the potential role of targeting the CXCL6/CXCR2 signaling axis in reducing chemoresistance in HCC. Thus, our study’s results are the foundation for future research in targeting this pathway.

## Figures and Tables

**Figure 1 biomedicines-11-02081-f001:**
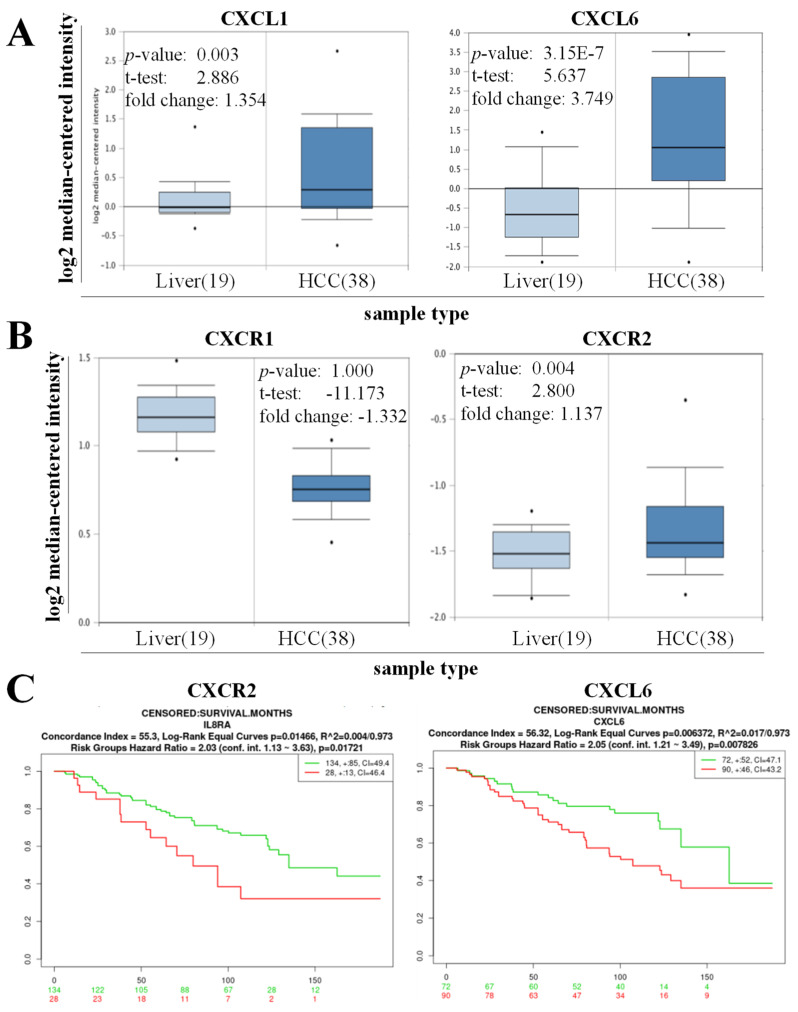
CXCL6 and CXCR2 are associated with hepatogenesis and reduced the overall survival of patients with hepatocellular carcinoma (HCC). (**A**) Mas Liver HCC cohort (*n* = 57) using the Oncomine platform demonstrated that patients with HCC exhibited higher expression of CXCL1 (1.35-fold, *p* = 0.003) and CXCL6 (3.75-fold, *p* = 3.15 × 10^−7^). (**B**) The expression of CXCR2 was suppressed among patients with HCC (−1.33-fold, *p* = 1.00), and the induced expression of CXCR2 was observed in the HCC samples compared to their normal tissue (1.137-fold, *p* = 0.004). (**C**) High expression of CXCR2 and CXCL6 was associated with worse overall survival in patients with liver cancer (Hoshida Golub Liver GSE10143 (*n* = 162): red curve denotes higher expression, and the green dot represents low expression).

**Figure 2 biomedicines-11-02081-f002:**
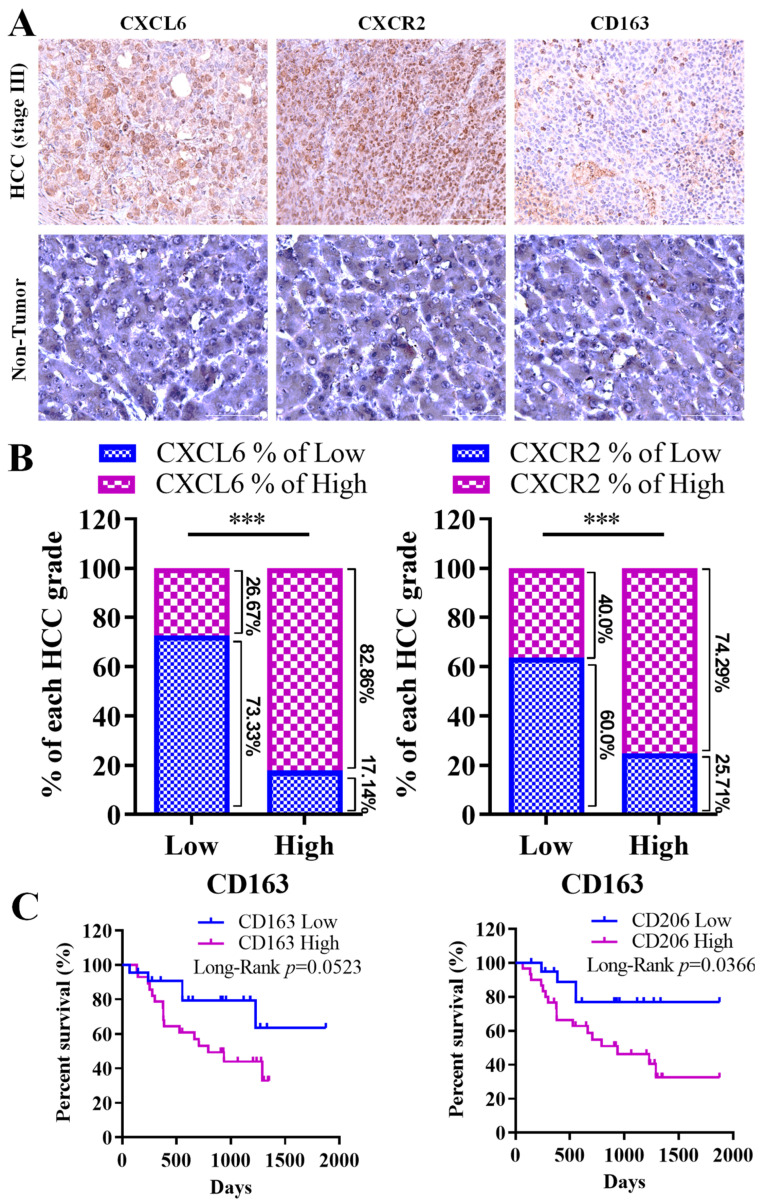
M2 macrophage tumoral infiltration, cancer progression, and poor prognosis in patients with hepatocellular carcinoma (HCC) are associated with the expression of CXCL6 and CXCR2. (**A**) Patients with HCC exhibited stronger expressions of CXCL6 and CXCR2 proteins than their normal peers. (**B**) Positive correlation between the expression of CXCL6, CXCR2, and the CD163 macrophage-associated antigen; the manifestation of CXCL6 and CXCR2 in patients presented as the percentage expression in CD163 high and low patient samples. (**C**) High expression of CD163 and CD206 negatively affected the overall survival of patients with HCC. ***, *p* < 0.001 high vs. low expression.

**Figure 3 biomedicines-11-02081-f003:**
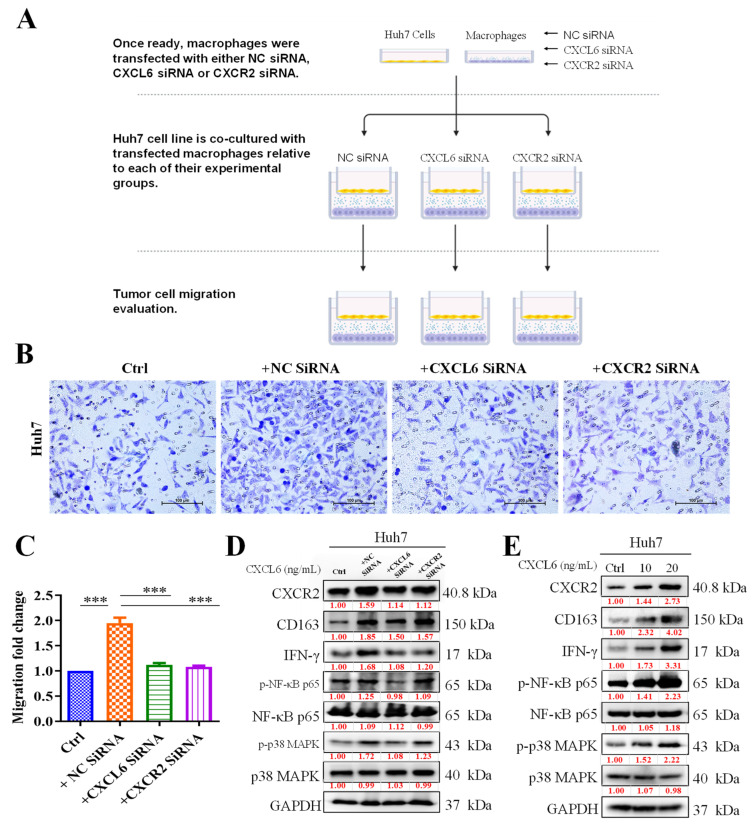
Activation of CXCR2/IFN-g/p38 MAPK/NF-κB signaling by tumor-associated macrophages (TAMs) was followed by increased cell proliferation and migration in hepatocellular carcinoma (HCC). (**A**) Schematic of experimental coculture methods for HCC and macrophages. (**B**) Huh7 cells cocultured with TAMs exhibited enhanced proliferation, which was reversed by the addition of TAMs transfected with si-CXCL6 or si-CXCR2. (**C**) Cell migration was also enhanced in cocultured TAM cells but suppressed in cells treated with si-CXCR2 or si-CXCL6. (**D**) Protein expression observed using the Western blot technique demonstrated that CXCL6/CXCR2 inhibition affected the expression of the IFN-g/p38 MAPK/NF-κB signaling pathway. (**E**) CXCL6 agonist addition increased the expression of signaling pathway. ***, *p* < 0.001, control vs. treatment.

**Figure 4 biomedicines-11-02081-f004:**
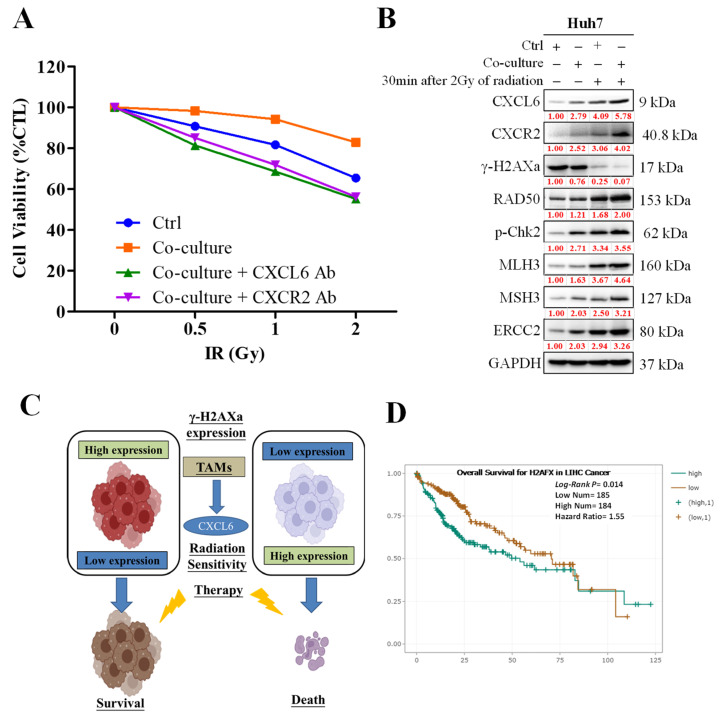
DNA damage repair system is enhanced by TAM CXCL6/CXCR2 signaling. (**A**) Coculturing with macrophages enhanced the viability of Huh7 cells exposed to radiotherapy. The viability of these cells was diminished by the addition of CXCL6 or CXCR2 antibodies. (**B**,**C**) Upregulated expression of DNA repair markers γ-H2AXa, Rad50, p-Chk2, MLH3, MSH3, and ERCC2 in cocultured macrophage cells compared with the control. (**D**) The Kaplan–Meier (KM) plot illustrated the correlation between the expression of a particular gene and patient survival rates.

**Figure 5 biomedicines-11-02081-f005:**
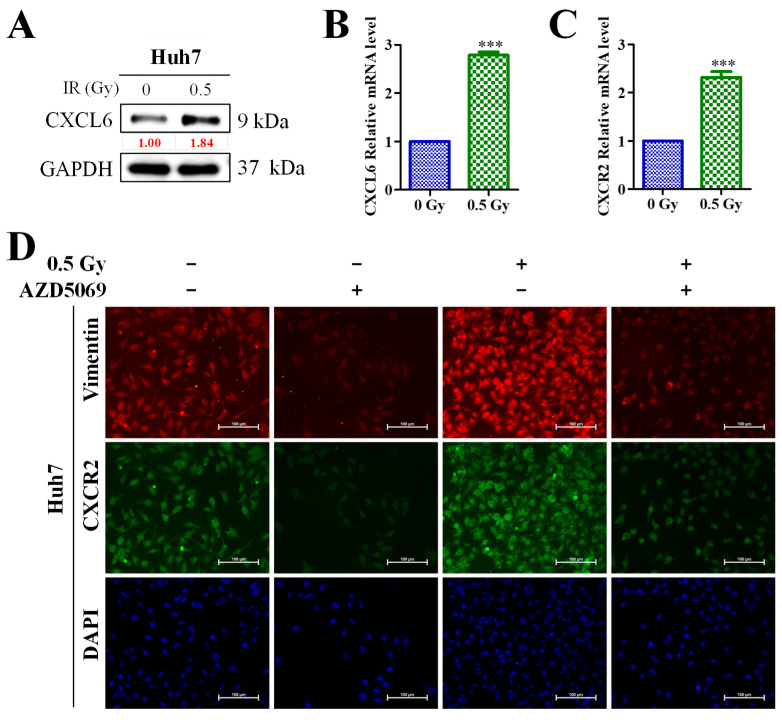
Low-dose (0.5-Gy) radiation-induced HCC cells epithelial–mesenchymal transition via the CXCL6/CXCR2 signaling pathway. (**A**) Low-dose radiation significantly upregulated expression levels of CXCL6 protein, (**B**) CXCL6 mRNA, and (**C**) CXCR2 mRNA. (**D**) CXCR2 inhibitor, AZD5069, significantly diminished the expression of vimentin after the application of low-dose radiotherapy. ***, *p* < 0.001, radiation (0.5 Gy) vs. control (0 Gy).

**Figure 6 biomedicines-11-02081-f006:**
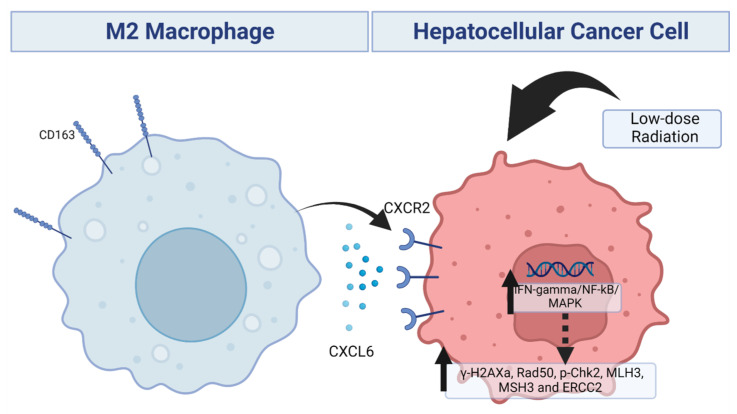
Schematic abstract. Illustration summarizing the concept of TAMs affecting HCC cell radioresistance via the CXCL6/CXCR2 signaling axis.

## Data Availability

The datasets used and analyzed in the current study are publicly accessible, as indicated in the manuscript.

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
