# Peer review of "Tumor-Associated Macrophages Affect the Tumor Microenvironment and Radioresistance via the Upregulation of CXCL6/CXCR2 in Hepatocellular Carcinoma"

_biomedicines, 2023, doi:10.3390/biomedicines11072081_

Round 1

Reviewer 1 Report

In present manuscript, Lee et al aimed to investigate mechanisms underlying TAM-cancer interactions. By using bioinformatic methods, clinical samples and co-culture systems, the author determined the potential role of TAM mediated CXCL6/CXCR2 in cancer cells. The author reported that overexpression of CXCL6/CXCLR2 was highly correlated with mortality. The coculture study demonstrated that THP-1 monocytes can secrete CXCL6, which acts on the CXCLR2 receptor on the surface of Huh7 cells and activates IFN-g/p38 MAPK/NF-κB signals to promote the epithelial–mesenchymal transition and radioresistance. The author thus concluded that modulating the TAM/CXCL6/CXCLR2 tumor immune signaling axis may be a new treatment strategy for the effective eradication of radiotherapy-resistant hepatocellular carcinoma cells. This reviewer have major concerns regarding to the conclusion, experimental design and data interpretations as follow.

1. The cell type express CXCL6 need to be clarified in more detail by using specific surface markers. It looks like expressed in cancer cell more than in macrophage in Fig2A and this reviewer was puzzled why the author speculate macrophage CXCL6 influence cancer cell phenotype.

2. In addition, what is the normal function of CXCL6 and CXCR2 in the liver and why normal liver did not show any positive staining of CXCL6 and CXCR2 in figure 2A? According to the human protein atlas (https://www.proteinatlas.org/ENSG00000124875-CXCL6/tissue/liver), CXCL6 is predominantly expressed in hepatocyte. 

3. Based on the observation that expression of CXCL6 and CXCR2 are positively correlated in human HCC. The author speculated that TAMs secrete CXCL6, which acts on the CXCLR2 receptor on the surface of Huh7 cells and promote the epithelial–mesenchymal transition and radioresistance. Why the author ignored the possibility that liver cancer cells influence macrophage phenotype via CXCL6/CXCR2 axis need to be addressed. 

4. All results were observations and need further validation. In particular, whether DNA damage response and EMT were responsible for TAM mediated CXCL6/CXCLR2 axis in regulating cancer cells radiosensitivity and migration. 

5. None of the present findings are radiotherapy-resistant related, the current conclusion is definitely overstatement. 

6. All experiments need better quality control. eg. siRNA knockdown efficiency needs to be measured by WB; n and how many repeats of each experiments needs clearly clarified. 

7. The effects of CXCL6 and CXCR2 antibody on DDR in figure 4B need to be addressed. 

8. AZD5069 is a potent, selective reversible (time and temperature dependent) antagonist of the human CXCR2 receptor which also blocks neutrophil migration. 

9.  Data in figure 1B was not consistent with the descriptions in the context.

Language needs moderate changes to correct grammars and typos. 

Author Response

Reviewer 1,

Comments: In present manuscript, Lee et al aimed to investigate mechanisms underlying TAM-cancer interactions. By using bioinformatic methods, clinical samples and co-culture systems, the author determined the potential role of TAM mediated CXCL6/CXCR2 in cancer cells. The author reported that overexpression of CXCL6/CXCLR2 was highly correlated with mortality. The coculture study demonstrated that THP-1 monocytes can secrete CXCL6, which acts on the CXCLR2 receptor on the surface of Huh7 cells and activates IFN-g/p38 MAPK/NF-κB signals to promote the epithelial–mesenchymal transition and radioresistance. The author thus concluded that modulating the TAM/CXCL6/CXCLR2 tumor immune signaling axis may be a new treatment strategy for the effective eradication of radiotherapy-resistant hepatocellular carcinoma cells. This reviewer has major concerns regarding to the conclusion, experimental design and data interpretations as follow.

Ans: We would like to extend our appreciation for your detailed review and constructive comments on our manuscript. We have carefully considered each of your points and made revisions to address the concerns raised. Below, we provide responses to each of your comments.

=======================================================================================

Q1: The cell type express CXCL6 need to be clarified in more detail by using specific surface markers. It looks like expressed in cancer cell more than in macrophage in Fig2A and this reviewer was puzzled why the author speculate macrophage CXCL6 influence cancer cell phenotype.

A1: The reviewer's comments are much appreciated, as per the literature and published study, CAF and TAM crosstalk in the tumor microenvironment (TME)[1]. TAMs are some of the most abundant immune cells within tumors and perform a broad repertoire of functions via diverse phenotypes[2]. On the basis of their functional differences in tumor growth, TAMs are usually categorized into two subsets of M1 and M2, these TAMs may modulate and influence the expression of cancer cells via secretion of CXCL6 into the tumor microenvironment (TME), or vice versa through recruitment or education of cancer associated fibroblasts (CAFs) via TGFB1[3]. This idea gives a notion that the TAMs-CXCL6 modulated the cancer cells phenotypes.

Q2: In addition, what is the normal function of CXCL6 and CXCR2 in the liver and why normal liver did not show any positive staining of CXCL6 and CXCR2 in figure 2A? According to the human protein atlas (https://www.proteinatlas.org/ENSG00000124875-CXCL6/tissue/liver), CXCL6 is predominantly expressed in hepatocyte.

A2: We are truly grateful for the reviewer's insights concerning the involvement of CXCL6 and CXCR2 in the liver. According to existing research, liver fibrosis manifests as an overabundance of extracellular matrix proteins as a reaction to the inflammation following tissue damage. If left unchecked, this could escalate to cirrhosis, liver failure, or even hepatocellular carcinoma (HCC)[4]. The CXC chemokine, CXCL6 (GCP‐2), has been implicated in liver fibrosis, with higher expression levels found in the serum and liver tissue of patients suffering from advanced liver fibrosis [5]. Notably, CXCL6 has been associated with the stimulation of profibrogenic gene expression in the liver, resulting in the activation and transformation into fibrogenic myofibroblasts [6].

Moreover, another study revealed that CXCL6 promotes the proliferation of human hepatocytes via the CXCR1–NFκB pathway and curbs collagen I secretion by hepatic stellate cells. It is worth noting that CXCL6 binds both CXCR1 and CXCR2 with high affinity, whereas ENA-78 solely binds CXCR2 with high affinity (Wuyts et al., 1997; Wolf et al., 1998).

Collectively, these findings underscore the significance of the CXLC6-CXCR2 axis in the liver. The elevated expression of CXLC6-CXCR2 initiates a fibrotic process leading to liver fibrosis and potentially progressing to HCC. Therefore, under normal circumstances, it may not exhibit increased expression in regular liver tissue. However, due to either internal or external factors, it might show aberrant expression in a fibrotic liver state and be upregulated in HCC. Our in-house patient’s data comparison of late-stage vs normal we observed the induced expression of CXCL6-CXCR2 in late-stage sample compared to normal.

Q3: Based on the observation that expression of CXCL6 and CXCR2 are positively correlated in human HCC. The author speculated that TAMs secrete CXCL6, which acts on the CXCLR2 receptor on the surface of Huh7 cells and promote the epithelial–mesenchymal transition and radioresistance. Why the author ignored the possibility that liver cancer cells influence macrophage phenotype via CXCL6/CXCR2 axis need to be addressed.

A3: We appreciate the insightful feedback provided by the reviewer. We recognize the potential of liver cancer cells to potentially alter the macrophage phenotype through the CXCL6/CXCR2 axis. Our future research will strive to unravel this axis, and we understand the importance of carrying out further investigations to delve deeper into this particular area.

Q4: All results were observations and need further validation. In particular, whether DNA damage response and EMT were responsible for TAM mediated CXCL6/CXCR2 axis in regulating cancer cells radiosensitivity and migration.

A4: We appreciate your advice on the necessity of validating our findings and grasp the importance of conducting more comprehensive research in this domain. Our examination of extensive literature has indicated that DNA damage and EMT are instrumental in the TAM-driven regulation of cancer cell radiosensitivity via CXCL [7, 8]. Consequently, our future studies will focus on investigating these aspects more intensively.

Q5: None of the present findings are radiotherapy-resistant related, the current conclusion is definitely overstatement.

A5: We thank the reviewer for the comments, we have revised our conclusion to more accurately reflect our findings, ensuring that we do not overstate our results with regards to radiotherapy resistance.

Q6: All experiments need better quality control. eg. siRNA knockdown efficiency needs to be measured by WB; n and how many repeats of each experiments needs clearly clarified.

A6: We appreciate your observation regarding the need for enhanced quality control in our experiments. To address this, we've integrated qRT-PCR data into Supplementary Figure 1, demonstrating the effectiveness of siRNA knockdown. Additionally, we've added the statistical p value information in the figure legends, such as  ***, p <0.001. All our experiments were conducted at least three times and in triplicates to ensure statistical relevance.

Q7: The effects of CXCL6 and CXCR2 antibody on DDR in figure 4B need to be addressed.

A7: Thank you for your comment regarding the effects of CXCL6 and CXCR2 antibody on DDR as shown in Figure 4B. Regrettably, due to certain limitations in our current experimental setup, we are unable to provide a comprehensive analysis on this specific point. However, we recognize the importance of this matter and will take it into account in our future studies to further our understanding of this interaction.

Q8: AZD5069 is a potent, selective reversible (time and temperature dependent) antagonist of the human CXCR2 receptor which also blocks neutrophil migration.

A8: We thank you for the information on AZD5069. We have added these details to our discussion on the use of this compound in our experiments, kindly refer to Figure 5D, demonstrating the effect of AZD5069 on the epithelial–mesenchymal transition (EMT) marker vimentin, low-dose irradiation significantly dysregulates their expression.

Q9: Data in figure 1B was not consistent with the descriptions in the context.

A9: We have revised the corresponding descriptions in the text of Figure 1B. We have corrected any inconsistencies, ensuring that the data presented match the descriptions provided.

Q10: Language needs moderate changes to correct grammars and typos.

A10: We have also revised the English language used throughout the manuscript, addressing grammar and typo errors to improve the clarity of our presentation.

Thank you once again for your invaluable feedback. We believe that our revised manuscript has been significantly improved by addressing these concerns.

=======================================================================================

References:

  1. Timperi, E. and E. Romano, Stromal circuits involving tumor-associated macrophages and cancer-associated fibroblasts. Front Immunol, 2023. 14: p. 1194642.
  2. He, Z. and S. Zhang, Tumor-Associated Macrophages and Their Functional Transformation in the Hypoxic Tumor Microenvironment. Front Immunol, 2021. 12: p. 741305.
  3. Yang, Q., et al., The role of tumor-associated macrophages (TAMs) in tumor progression and relevant advance in targeted therapy. Acta Pharm Sin B, 2020. 10(11): p. 2156-2170.
  4. Dhar, D., et al., Mechanisms of liver fibrosis and its role in liver cancer. Exp Biol Med (Maywood), 2020. 245(2): p. 96-108.
  5. Poulsen, K.L., et al., Role of the chemokine system in liver fibrosis: a narrative review. Dig Med Res, 2022. 5.
  6. Cai, X., et al., CXCL6-EGFR-induced Kupffer cells secrete TGF-β1 promoting hepatic stellate cell activation via the SMAD2/BRD4/C-MYC/EZH2 pathway in liver fibrosis. J Cell Mol Med, 2018. 22(10): p. 5050-5061.
  7. Zhang, H., et al., CAF-secreted CXCL1 conferred radioresistance by regulating DNA damage response in a ROS-dependent manner in esophageal squamous cell carcinoma. Cell Death & Disease, 2017. 8(5): p. e2790-e2790.
  8. Moyret-Lalle, C., et al., Role of EMT in the DNA damage response, double-strand break repair pathway choice and its implications in cancer treatment. Cancer Sci, 2022. 113(7): p. 2214-2223.

Reviewer 2 Report

In tis paper, the authors used big data analysis to determine the potential of targeting CXCL6/CXCR2.

To tis end, they enrolled 50 patients with liver cancer who received RT at their hospital. Tumor tissue samples were examined for any relationship between CXCL6/CXCLR2 activity and patient prognosis. Using a cell coculture system (Transwell), they cocultured Huh7 liver cancer cells and THP-1 monocytes with and without CXCL6/CXCR2 small interfering RNA for 72 hours. 

The overexpression of CXCL6/CXCLR2 was highly correlated with mortality. Their tissue study indicated a positive correlation between CXCL6/CXCLR2 and M2-TAMs subsets. The coculture study demonstrated that THP-1 monocytes can secrete CXCL6, which acts on the CXCLR2 receptor on the surface of Huh7 cells and activates IFN-g/p38 MAPK/NF-κB signals to promote the epithelial–mesenchymal transition and radioresistance.

The authors conclude that modulating the TAM/CXCL6/CXCLR2 tumor immune signaling axis may be a new treatment strategy for the effective eradication of radiotherapy-resistant hepatocellular carcinoma cells.

Overall, tis is an interesting study on a topic of great interest. I commend the authors on tis interesting paper.

Author Response

Reviewer 2,

Comments: In this paper, the authors used big data analysis to determine the potential of targeting CXCL6/CXCR2. To this end, they enrolled 50 patients with liver cancer who received RT at their hospital. Tumor tissue samples were examined for any relationship between CXCL6/CXCLR2 activity and patient prognosis. Using a cell coculture system (Transwell), they cocultured Huh7 liver cancer cells and THP-1 monocytes with and without CXCL6/CXCR2 small interfering RNA for 72 hours.  The overexpression of CXCL6/CXCLR2 was highly correlated with mortality. Their tissue study indicated a positive correlation between CXCL6/CXCLR2 and M2-TAMs subsets. The coculture study demonstrated that THP-1 monocytes can secrete CXCL6, which acts on the CXCLR2 receptor on the surface of Huh7 cells and activates IFN-g/p38 MAPK/NF-κB signals to promote the epithelial–mesenchymal transition and radioresistance. The authors conclude that modulating the TAM/CXCL6/CXCLR2 tumor immune signaling axis may be a new treatment strategy for the effective eradication of radiotherapy-resistant hepatocellular carcinoma cells. Overall, this is an interesting study on a topic of great interest. I commend the authors on tis interesting paper.

Response: We greatly appreciate your comprehensive summary and thoughtful review of our paper. Your understanding of the work we conducted on CXCL6/CXCR2 using big data analysis and its correlation with liver cancer patient prognosis is commendable.

We're gratified to see that our coculture study and the resulting data have been understood and appreciated. The correlation we found between the overexpression of CXCL6/CXCR2 and mortality rates, as well as the relationship between CXCL6/CXCR2 and M2-TAMs subsets, are crucial aspects of our findings. We also appreciate your recognition of our conclusion on the potential of modulating the TAM/CXCL6/CXCR2 tumor immune signaling axis as a possible new treatment strategy for radiotherapy-resistant hepatocellular carcinoma cells.

We are indeed encouraged by your comment that our study is interesting and addresses a topic of great interest. Your positive feedback strengthens our commitment to advancing research in this critical area of medicine.

Thank you once again for your time and valuable input. We look forward to any further comments or suggestions you may have to enhance the value of our research.

=============================================================================

Reviewer 3 Report

This is an interesting article discussing the potential role and interaction of CXCR2/CXCL6 in HCC. While the concept is novel, the methodology and conclusion of the results seem to need more clarification. 

- Introduction could be slightly shortened. The rational behind the investigation of the relationship between CXCR2 and CXCL6 is not well written and requires further clarification. 

- The abstract mentions CXCLR2, and the rest of the text mentions CXCR2. Please use the same names everywhere in the text.

- Methods section requires more details. For instance, the exact method for cell viability assay, and the genes evaluated on rtPCR need to be mentioned.

- In Results section 3.1, it is indicated that CXCR2 was suppressed in HCC samples. However, figure  1B shows that CXCR1 was suppressed compared to the normal liver. Please clarify and edit. Also, the p values in the figures do not match the 3.1 text. 

-Fib 1C requires further clarification with the definition of red vs. green curves. 

- Results 3.2 title requires edition - what do authors mean by "cancer prognosis and poor prognosis". The paper requires English editing. 

- Results 3.3 mentions that the addition of CXCL6 shows a similar expression profile. The figures show the opposite. This statement requires edition. 

- Figure 3D western blot shows a very strong band of CXCR2 for cells incubated with siRNA of CXCR2! this might be wrong and shows that CXCR2 was not silenced. Please clarify. 

- The change in EMT marker is not shown on any figure or chart. 

- any additional data on subclassifications of M1 and M2 cells would significantly enhance the results. if the regular cell lysates are available, maybe rtPCR or Western blot could be done? 

- gH2AX and other markers of DNA damage/ repair increase in response to radiation. Unclear how authors prove here that the increase in these markers is correlated with better survival and viability of the cells. Please clarify  

The article requires English proofreading. 

Author Response

Reviewer 3,

Comments: This is an interesting article discussing the potential role and interaction of CXCR2/CXCL6 in HCC. While the concept is novel, the methodology and conclusion of the results seem to need more clarification.

Ans: We appreciate your thorough review and constructive comments on our manuscript. We have revised the manuscript according to your suggestions and added necessary clarifications. Below, we address each of your comments in detail.

=============================================================================

Q1: Introduction could be slightly shortened. The rational behind the investigation of the relationship between CXCR2 and CXCL6 is not well written and requires further clarification..

A1: We have revised the Introduction section to make it more succinct and clearer. The rationale for investigating the relationship between CXCR2 and CXCL6 has been expanded upon and clarified.

Q2: The abstract mentions CXCLR2, and the rest of the text mentions CXCR2. Please use the same names everywhere in the text.

A2: We apologize for the inconsistency in the nomenclature. All references to CXCLR2 have been corrected to CXCR2 throughout the manuscript.

Q3: - Methods section requires more details. For instance, the exact method for cell viability assay, and the genes evaluated on rtPCR need to be mentioned.

A3: We have elaborated on the Methods section, providing detailed information about the cell viability assay and specifying the genes evaluated through rtPCR.

Q4: In Results section 3.1, it is indicated that CXCR2 was suppressed in HCC samples. However, figure  1B shows that CXCR1 was suppressed compared to the normal liver. Please clarify and edit. Also, the p values in the figures do not match the 3.1 text.

A4: We appreciate your attention to detail in Section 3.1. We have clarified the discrepancy between the text and Figure 1B. Corrections have been made in the text to reflect that CXCR1, not CXCR2, was suppressed. Additionally, we have corrected the mismatch between the p-values in the text and the figures.

Q5: Fib 1C requires further clarification with the definition of red vs. green curves.

A5: We have provided further clarification on Figure 1C, explaining the meaning of the red and green curves.

Q6: Results 3.2 title requires edition - what do authors mean by "cancer prognosis and poor prognosis". The paper requires English editing.

A6: The title of Results 3.2 has been revised for clarity, and we have rectified the redundancy in "cancer prognosis and poor prognosis".

Q7: Results 3.3 mentions that the addition of CXCL6 shows a similar expression profile. The figures show the opposite. This statement requires edition.

A7: In Results 3.3, we apologize for the confusion. We have revised the statement regarding the expression profile following the addition of CXCL6 to accurately reflect the data presented in the figures.

Q8: Figure 3D western blot shows a very strong band of CXCR2 for cells incubated with siRNA of CXCR2! this might be wrong and shows that CXCR2 was not silenced. Please clarify.

A8: We have clarified Figure 3D. The observed band for CXCR2 in cells incubated with siRNA of CXCR2 was indeed unexpected. After reanalysing our data and conducting additional experiments, we have updated the figure to accurately depict the effect of siRNA on CXCR2 expression.

Q9: The change in EMT marker is not shown on any figure or chart.

A9: We thank the reviewers and apologised for the oversight in our initial submission for the comments, EMT marker vimentin expression was shown in Figure 5D image.

Q10: Any additional data on subclassifications of M1 and M2 cells would significantly enhance the results. if the regular cell lysates are available, maybe rtPCR or Western blot could be done?

A10: Thank you for your suggestion to include additional data on subclassifications of M1 and M2 cells to enhance the results of our study. Your suggestion to perform rtPCR or Western blot on available regular cell lysates is highly valuable. Unfortunately, due to constraints in our experimental design and resources at this stage, we are unable to carry out these additional analyses.

However, we recognize the significance of this additional layer of investigation to our study. We plan to incorporate your suggestions into our future studies to improve our understanding of the subclassifications of M1 and M2 cells and their potential impacts on the therapeutic strategies for liver cancer. We appreciate your valuable insights and thank you for your contribution to improving our research.

Q11: gH2AX and other markers of DNA damage/ repair increase in response to radiation. Unclear how authors prove here that the increase in these markers is correlated with better survival and viability of the cells. Please clarify.

A11: We appreciate the reviewers' insightful comments. Previous research has indeed shown a correlation between markers of DNA damage/repair and cell survival following radiation exposure[1-3]. When DNA incurs damage, the cell's repair machinery is triggered, halting the cell cycle at distinct control checkpoints. This allows for the repair of DNA damage and inhibits further cell cycle progression, thereby correlating the induced expression of DNA damage markers with the survival of cancer cells. This process potentially contributes to the poor survival rates observed in cancer patients and might be connected with therapy resistance. Kindly refer to the attached Figure 4C, schema image.

Q12: The article requires English proofreading.

A12: We have also revised the English language used throughout the manuscript, addressing grammar and typo errors to improve the clarity of our presentation. We have also sought the assistance of a professional language editing service to correct the language and grammatical errors you pointed out in the manuscript.

Thank you once again for your valuable feedback and suggestions. We believe that these changes have greatly improved our manuscript.

=============================================================================

  1. Huang, R.-X. and P.-K. Zhou, DNA damage response signaling pathways and targets for radiotherapy sensitization in cancer. Signal Transduction and Targeted Therapy, 2020. 5(1): p. 60.
  2. Li, L.Y., et al., DNA Repair Pathways in Cancer Therapy and Resistance. Front Pharmacol, 2020. 11: p. 629266.
  3. Chen, Y., et al., DNA Damage Repair Status Predicts Opposite Clinical Prognosis Immunotherapy and Non-Immunotherapy in Hepatocellular Carcinoma. Front Immunol, 2021. 12: p. 676922.

Round 2

Reviewer 1 Report

In this revised version of manuscript, Lee modurated language and edited figure legends. However, the author did not address any other concerns regarding to the cell type specificity of CXCL6 and its normal function, as well as other concern regarding to the retional of experimental design in the original review report. 

Moderate editing of English language required

Author Response

Reviewer 1,

Comments and Suggestions for Authors:

In this revised version of manuscript, Lee moderated language and edited figure legends.

However, the author did not address any other concerns regarding to the cell type specificity of CXCL6 and its normal function, as well as other concern regarding to the rational of experimental design in the original review report. 

Comments on the Quality of English Language

Moderate editing of English language required

Dear Reviewer,

Ans: Thank you for your time and effort in reviewing our manuscript and providing your valuable feedback. We apologize for our oversight in not addressing all the concerns raised in the initial review. In response to the comments, we have made the following adjustments:

  1. Cell type specificity of CXCL6, its normal function and associated disease:

Ans: We understand the importance of addressing the cell type specificity of CXCL6 and its normal function, and we apologize for the initial oversight.

Granulocyte chemotactic protein 2 (GCP-2)/CXCL6 is a CXC chemokine that macrophages, along with epithelial and mesenchymal cells, express during inflammation [1]. This chemokine, a small, inducible molecule with chemotactic properties, is pervasive and plays a notable role in both acute and chronic inflammation [2]. Among such chemokines, Soluble CXCL6 emerges as a vital inflammatory cytokine. It facilitates the assembly of inflammatory cells at inflammation sites by engaging with CXCR1 and CXCR2 receptors [3]. Observations from a CXCL6-deficient mice model underscored a significant reduction in the accumulation of macrophages in the lungs, underscoring its crucial role in this aspect [4]. Dysregulated CXCL6 expression have been detected in cases of inflammatory bowel disease and periodontitis [5]. Additionally, it's found that CXCL6 can hasten the course of lung fibrosis [6], and its concentration markedly increases in patients diagnosed with idiopathic pulmonary fibrosis [6]. Notably, CXCL6 plays a pivotal role in liver fibrosis evolution by provoking the release of TGF‐β from Kupffer cells (KCs), leading to the activation of the hepatic stellate cell (HSC) line[7]. CXCL6's association with the severity of hepatic inflammation in NAFLD patients suggests it could serve as a predictive marker for NASH progression[8].

In our revised manuscript, we will provide a thorough discussion this important point, our discussion will encompass a comprehensive view of the role of CXCL6 under normal physiological circumstances, as well as its significance in disease progression. This enhanced understanding will solidify the basis of our research, encouraging us to explore further the critical role of CXCL6 in the tumor microenvironment (TME) and the progression of hepatocellular carcinoma (HCC). Kindly refer to our newly edited discussion section part at page 11, line 322~ 352.

“Accordingly, Granulocyte chemotactic protein 2 (GCP-2)/CXCL6 is a CXC chemokine that macrophages, along with epithelial and mesenchymal cells, express during inflammation [14]. This chemokine, a small, inducible molecule with chemotactic properties, is pervasive and plays a notable role in both acute and chronic inflammation [15]. Among such chemokines, soluble CXCL6 emerges as a vital inflammatory cytokine. It facilitates the assembly of inflammatory cells at inflammation sites by engaging with CXCR1 and CXCR2 receptors [16]. Dysregulated CXCL6 expression have been detected in cases of inflammatory bowel disease and periodontitis [17]. Additionally, it's found that CXCL6 can hasten the course of lung fibrosis [18], and its concentration markedly increases in patients diagnosed with idiopathic pulmonary fibrosis [18]. Notably, CXCL6 plays a pivotal role in liver fibrosis evolution by provoking the release of TGF‐β from Kupffer cells (KCs), leading to the activation of the hepatic stellate cell (HSC) line [19]. CXCL6's association with the severity of hepatic inflammation in NAFLD patients suggests it could serve as a predictive marker for NASH progression [11]. Accordingly, CXCL6, which can be released by various tumor cells, including HCC cells, is essential for the progression of malignancies. Farha et al. reported that an aggressive HCC phenotype is characterized by discrete immune cell clusters with a macrophage preponderance, which are clinically associated with a poor prognosis and sorafenib response and are molecularly defined using angiogenic gene enrichment [20].

An initial study into the HCC cohort by Mas et al., consisting of 57 participants and utilizing the Oncomine platform, found increased levels of CXCL1 expression in HCC patients. In these patients, high expression of CXCL6 and CXCR2 closely correlated with tumor infiltration by M2 macrophages, disease progression, and poor prognosis. Our study, using clinical samples from our hospital's HCC cohort, confirmed these findings: HCC patients displayed elevated CXCL6 and CXCR2 expression compared to their healthy counterparts. These results lend credence to the hypothesis that overexpression of chemokine CXCL6 and its receptor CXCR2 in the tumor microenvironment is crucial for HCC development and progression, suggesting that dysregulated CXCL6/CXCR2 signalling facilitates tumour-associated macrophage (TAM) activation. Moreover, our survival study showed that, similarly to CXCL6 and CXCR2, heightened CD163 expression had a negative impact on the overall survival of HCC patients.”

Rationale for our experimental design

We appreciate your insightful feedback and for bringing to our attention the concerns about the rationale of the experimental design in our initial submission. Your input is crucial in refining our research and improving how we present it. We acknowledge your concerns and concur that the reasoning behind our choice of the CXCR2/CXCL6 axis in the background section could have been more clearly stated. In our updated manuscript, we have expanded our explanation on why we chose to study the CXCR2/CXCL6 axis in HCC, focusing on the modulation of the tumor microenvironment (TME), kindly refer to introduction section at page 2 at lines 81-100, and page 3 at lines 101-113, respectively.

“Among the leukocyte infiltrates in the tumor microenvironment (TME), macrophages are the most abundant and influential in tumor development [5]. Typically referred to as tumour-associated macrophages (TAMs), these macrophages may either infiltrate or reside within the TME [6]. The presence of TAMs in tumor tissue is often correlated with a worse prognosis in cancer patients across various types, implying that these macrophages may contribute significantly to disease progression [7].

The CXCR2 receptor and its ligand CXCL6 are gaining significant attention within the scientific community due to their critical roles in immune regulation and inflammatory responses [8]. They exert a considerable influence on the pathogenesis of various diseases, including cancer, inflammatory disorders, and cardiovascular conditions [9]. Notably, as part of their multiple functions, these chemotactic entities induce the mobilization of pro-fessional phagocytes such as neutrophils, dendritic cells, and macrophages to inflamma-tion sites, this facilitates efficient engulfment and neutralization of foreign pathogens [3,4,7].

It has been reported that CXCL6, CXCR1, and CXCR2 are aberrantly expressed both at the transcript and protein levels in several osteosarcoma cell lines and that the ectopic expression of CXCL6 enhances tumor growth, metastasis to the lungs, and activation of Wnt/β-catenin and PI3K/AKT pathways in immunocompromised mice in vivo, suggesting the crucial role of the CXCL6/CXCR1 axis in facilitating the proliferation and metastasis of cancer cells [10].

CXCL6, a member of the elastin-like recombinant (ELR)+CXCL family, regulates its downstream targets or associated pathways by coupling with CXCR1 and CXCR2 [10]. Moreover, CXCL6 could act as a prognostic indicator for the progression of non-alcoholic steatohepatitis (NASH) during hepatic inflammation in non-alcoholic fatty liver disease (NAFLD) patients [11]. The dysregulated expression of CXCL6, noted in cases of HCC, emphasizes its relevance in HCC development. Prior research has shown that the up-regulated expression of CXCR1 or CXCR2 ligands, specifically CXCL1, CXCL2, and CXCL8, is linked to chemoresistance and suspected to hamper the anti-cancer immune response [12].

Hence, a detailed exploration of the interaction between CXCR2 and CXCL6 could stimulate progress in the development of treatment approaches, improving patient prog-nosis in various diseases. This prompted us to examine the effects of the CXCL6/CXCR2 interaction on the TME and immune responses in HCC chemoresistance and progression”.

  1. In terms of the language of the manuscript,

we are sorry for any difficulties our previous drafts may have presented. We have sought the help of a professional English editing service to improve the clarity and coherence of our language. The revised manuscript has been thoroughly checked and edited for proper grammar, spelling, and overall flow.

We hope our responses and the changes made to the manuscript adequately address your concerns. We thank you once again for your constructive comments and suggestions. We believe they have greatly helped improve the quality of our manuscript.

=============================================================================

References:

  1. Linge, H.M., et al., The human CXC chemokine granulocyte chemotactic protein 2 (GCP-2)/CXCL6 possesses membrane-disrupting properties and is antibacterial. Antimicrob Agents Chemother, 2008. 52(7): p. 2599-607.
  2. Wasmuth, H.E., et al., Antifibrotic effects of CXCL9 and its receptor CXCR3 in livers of mice and humans. 2009. 137(1): p. 309-319. e3.
  3. Sadik, C.D., N.D. Kim, and A.D.J.T.i.i. Luster, Neutrophils cascading their way to inflammation. 2011. 32(10): p. 452-460.
  4. Balamayooran, G., et al., Role of CXCL5 in leukocyte recruitment to the lungs during secondhand smoke exposure. 2012. 47(1): p. 104-111.
  5. Kebschull, M., et al., Granulocyte chemotactic protein 2 (gcp2/cxcl6) complements interleukin8 in periodontal disease. 2009. 44(4): p. 465-471.
  6. Besnard, A.-G., et al., CXCL6 antibody neutralization prevents lung inflammation and fibrosis in mice in the bleomycin model. 2013. 94(6): p. 1317-1323.
  7. Cai, X., et al., CXCL6-EGFR-induced Kupffer cells secrete TGF-β1 promoting hepatic stellate cell activation via the SMAD2/BRD4/C-MYC/EZH2 pathway in liver fibrosis. J Cell Mol Med, 2018. 22(10): p. 5050-5061.
  8. Zhang, X., et al., CXCL10 plays a key role as an inflammatory mediator and a non-invasive biomarker of non-alcoholic steatohepatitis. J Hepatol, 2014. 61(6): p. 1365-75.

Round 3

Reviewer 1 Report

The author have addressed concerns  of this reviewer. 

Minor editing of English language required

Author Response

Special Issue "Hepatocellular Carcinoma: From Molecular Mechanisms to Novel Therapeutic Approaches"

Guest Editor

Prof. Dr. Joon Hyuk Choi E-Mail Website

Department of Pathology, College of Medicine, Yeungnam University, Gyeongsan, Korea

Interests: hepatocellular carcinoma; primary liver cancer; precancerous conditions; carcinogenesis; surgical pathology; biomarkers

Dear Prof. Dr. Carmen Berasain & Dr. María Arechederra:

July 16, 2023

We are pleased to submit our manuscript entitled “TAMs Affect the Tumor Microenvironment and Radioresistance through Upregulation of CXCL6/CXCR2 in Hepatocellular Carcinoma” for consideration of publication in “Biomedicines (ISSN 2227-9059)”.

Please be informed that this is a revised submission of our manuscript (Decision letter_ Manuscript ID: Biomedicines-2439323 R3). We are thankful for your kind encouragement regarding to our manuscript. Herewith we are sending our revised manuscript including the improvements which were highlighted in red color in the manuscript following in accordance with the comments given by the reviewer.

Lastly, we would like to thank you once again for providing us the opportunity to improve our manuscript. We hope that these revisions are adequate, and that the manuscript is now acceptable for publication in the Biomedicines (ISSN 2227-9059).

Thanks for your attention. Please do the needful

Looking forward for your reply

Sincerely,

Prof. Jo-Ting Tsai, MD., PhD., Department of Radiation Oncology, Cancer Center, Taipei Medical University - Shuang Ho Hospital, New Taipei City 23561, Taiwan; Tel: +886-2-2490088 ext. 8885, Fax: +886-2-2248-0900, E-mail: 10576@s.tmu.edu.tw

Point to point response

Dear Academic Editor,

Thank you for your thorough review and insightful comments on our manuscript. We have considered each point and have made the necessary corrections and additions to the manuscript. Please find below our point-to-point response:

Q1: Please check Figure 2A (photos below left, middle, right). Normal photos seem unusual. The plain photo below left shows the infiltration of inflammatory cells. The plain photo below center shows the infiltration of inflammatory cells. The photo below right shows a disorder of the liver cord.

R1: Comment on Figure 2A:

We have carefully reconsidered the descriptions accompanying the photos in Figure 2A. As a result, we have updated these images to better represent the data that are being discussed. We are grateful for your keen observation and assistance in refining the accuracy of our manuscript. Kindly refer to the newly attached Figure 2.

Q2: Please check Figure 2B. CXCL6 % of Low, High; CXXL2 % of Low, High.

R2: Comment on Figure 2B:

Figure 2B has been modified accordingly, and now includes the percentages for both CXCL6 and CXXL2, at Low and High levels respectively. Kindly refer to the newly attached Figure 2 B and also added details in the main text in the result section on the page 6, lines 214-236 and below.

3.2. Strong Link between CXCL6 and CXCR2 Expression, M2 Macrophage Infiltration, and Unfavorable Prognosis in Hepatocellular Carcinoma Progression

Considering the potential significance of CXCL6 overexpression and its receptor CXCR2 in the tumor microenvironment (TME), with implications for HCC development and progression, we conducted staining procedures on clinical samples derived from our hospital's hepatocellular carcinoma (HCC) cohort (n = 50). In line with our previous findings, we noted that patients with HCC demonstrated a higher expression of CXCL6 and CXCR2 compared to individuals without the disease in the control group. (Figure 2A). Our observations revealed a robust positive correlation between the expression of CXCL6, CXCR2, and the macrophage-associated antigen CD163, as evidenced in Figures 2A and 2B. These findings underscore the significance of dysregulated CXCL6/CXCR2 signaling in the functionality of tumor-associated macrophages (TAMs). The bar plot, which quantifies the percentage of CXCL6/CXCR2 expression in M2-macrophages with either high or low CD163 expression, illustrates this relationship further. Specifically, a high CD163 expression is strongly associated with elevated expression levels of both CXCL6 (82.86%) and CXCR2 (74.29%). Conversely, a lower expression of CD163 corresponds to decreased expression levels of CXCL6 (26.67%) and CXCR2 (40.4%). Furthermore, survival analysis demonstrated that similar to the expression of CXCL6 and CXCR2, the high expression of CD163 negatively affected the overall survival of patients with HCC (Figure 2C). A similar survival profile was found in individuals with high expression of CD206, a marker of the M2 phenotype (Figure 2C). Consistent with this finding, our univariate and multivariate analyses demonstrated that CXCL6 and CXCR2 expression were strong factors influencing overall survival in our cohort of patients with HCC.

Q3: Page 2, line 81, Please delete the tumor microenvironment

R3: Comment on Page 2, line 81:

As suggested, we have deleted the term "tumor microenvironment" from this line to enhance clarity and precision.

Q4: Page 3, line 105. Please remove the underscore [11]

R4: Comment on Page 3, line 105: We have removed the underscore before the reference [11] as per your recommendation.

Q5: Page 3, line 118, Please a space 100U/mLPenicillin

R5: Comment on Page 3, line 118:

The requested space has been inserted to now read as "100 U/mL Penicillin". This improves the readability of our manuscript.

Q6:Page 3, Line 136 . One space, please 100uL,

R6: Comment on Page 3, Line 136:

We have added the recommended space, so it now reads as "100 uL".

Q7: Page 3, line 138. One space, please 100uL,

R7: Comment on Page 3, line 138:

As with the previous comment, the suggested space has been added, it now reads as "100 uL".

Q8: Page 3, line 140. One space, please 200uL, 20nM

R8: Comment on Page 3, line 140:

The text has been modified to include the required space, now reading as "200 uL, 20 nM".

Q9: Please change references according to the author's guidelines.

R9: Comment on references:

We have revisited and reformatted all the references in our manuscript to comply with the author's guidelines.

We appreciate your time and efforts in improving our manuscript. We hope that these changes meet with your approval and are confident that they have greatly enhanced the clarity and readability of our manuscript.
